# Distinct stages of synapse elimination are induced by burst firing of CA1 neurons and differentially require MEF2A/D

**Chia-Wei Chang[†], Julia R Wilkerson[†], Carly F Hale, Jay R Gibson, Kimberly M Huber***

Department of Neuroscience, University of Texas Southwestern Medical Center, Dallas, United States

**Abstract** Experience and activity refine cortical circuits through synapse elimination, but little is known about the activity patterns and downstream molecular mechanisms that mediate this process. We used optogenetics to drive individual mouse CA1 hippocampal neurons to fire in theta frequency bursts to understand how cell autonomous, postsynaptic activity leads to synapse elimination. Brief (1 hr) periods of postsynaptic bursting selectively depressed AMPA receptor (R) synaptic transmission, or silenced excitatory synapses, whereas more prolonged (24 hr) firing depressed both AMPAR and NMDAR EPSCs and eliminated spines, indicative of a synapse elimination. Both synapse silencing and elimination required de novo transcription, but only silencing required the activity-dependent transcription factors MEF2A/D. Burst firing induced MEF2A/D-dependent induction of the target gene *Arc* which contributed to synapse silencing and elimination. This work reveals new and distinct forms of activity and transcription-dependent synapse depression and suggests that these processes can occur independently.

DOI: https://doi.org/10.7554/eLife.26278.001

**\*For correspondence:**
kimberly.huber@utsouthwestern.edu

[†]These authors contributed equally to this work

## Introduction

Sensory experience and learning refine cortical circuits through activity-dependent stabilization and elimination of excitatory synaptic connections (*Hua and Smith, 2004*; *Zuo et al., 2005*; *Fu and Zuo, 2011*). Surprisingly little is known of the molecular mechanisms by which experience and neural activity patterns lead to synapse elimination in cortical neurons. Importantly, deficits in experience-dependent synapse elimination are associated with neurodevelopmental disorders, such as Fragile X Syndrome, autism and schizophrenia (*Pfeiffer et al., 2010*; *Patel et al., 2014*; *Tang et al., 2014*; *Nagaoka et al., 2016*; *Sekar et al., 2016*). Determining the mechanisms and roles of disease-linked genes in synapse elimination will be key to understanding and treating these disorders. Activity- and experience-regulated transcriptional control is a candidate mechanism to transduce activity signals into long-term changes in circuit connectivity (*Flavell and Greenberg, 2008*; *West and Greenberg, 2011*). In support of this idea, the MEF2 (Myocyte-Enhancer Factor 2) family of activity-dependent transcription factors has been implicated in functional and structural elimination of excitatory synapses (*Flavell et al., 2006*). All four MEF2 family members (MEF2A-D) are expressed in the brain. MEF2A and MEF2D are highly expressed in hippocampal CA1 neurons and MEF2C is expressed more in the dentate gyrus and neocortex (*Lyons et al., 1995*). MEF2C and MEF2D as well as MEF2 target genes, are implicated in human autism, intellectual disability, schizophrenia, Alzheimer's disease and/or epilepsy (*Rocha et al., 2016*). In mice, deletion of *Mef2a,c,* and/or *d* leads to behaviors relevant to autism and drug addiction and alterations in learning and memory (*Pulipparacharuvil et al., 2008*; *Cole et al., 2012*; *Harrington et al., 2016*). Regulation of activity-

dependent circuit refinement by MEF2 may contribute to these behaviors and brain disorders in humans.

Most evidence linking MEF2 activation with synapse elimination comes from studies utilizing a constitutive, transcriptionally active MEF2, constructed by fusing the MADS/MEF2 DNA binding domains of MEF2C to the viral transcriptional activator VP16. MEF2-VP16 expression causes rapid elimination of excitatory synapses in cultured hippocampal neurons (*Flavell et al., 2006*; *Pfeiffer et al., 2010*; *Wilkerson et al., 2014*) and in vivo (*Barbosa et al., 2008*; *Hu et al., 2010*). In support of a role for endogenous MEF2 genes in synapse elimination, MEF2C knockout mice display enhanced spine number and excitatory synaptic function of dentate gyrus granule cells (*Barbosa et al., 2008*) and knockdown of MEF2A/D in hippocampal cultures increases excitatory synapse markers, an effect that relies on activity of the cultures (*Flavell et al., 2006*). However, MEF2A/D knockdown does not always lead to increases in synapse number (*Akhtar et al., 2012*; *Elmer et al., 2013*) suggesting there may be specific activity patterns or other factors necessary to observe MEF2A/D dependent synaptic plasticity. Furthermore, nothing is known of the physiological activity patterns that regulate MEF2 transcriptional activity in neurons or whether such patterns eliminate synapses through MEF2 transcriptional activation of specific gene targets.

To address these questions, we used optogenetics to drive firing of individual CA1 neurons in specific firing patterns. Blue light driven firing of postsynaptic neurons in 3 Hz bursts (patterned photostimulation; 1 hr; brief PPS;) depressed AMPA receptor (R) mediated synaptic transmission, but not that mediated by NMDARs, thus silencing excitatory synapses. In contrast, increasing the duration of PPS to 24 hr, depressed both AMPA and NMDAR synaptic transmission and eliminated dendritic spines (*Goold and Nicoll, 2010*). Surprisingly, MEF2A/D was necessary for activity-induced silencing of excitatory synapses, but not functional synapse elimination, revealing an unexpected role for endogenous MEF2A/D genes in synaptic plasticity. Brief PPS induced a MEF2A/D-dependent induction of the immediate-early gene *Arc*, previously implicated in the selective endocytosis of AMPARs, which was necessary, but not sufficient, for synapse silencing and contributed to synapse elimination. These results suggest that there are mechanistically distinct steps and multiple MEF2A/D-dependent target genes by which patterned firing of neurons leads to synapse elimination.

## Results

### Cell autonomous bursts of theta frequency postsynaptic action potentials induce a long-term depression of excitatory synaptic transmission

MEF2 is an activity-dependent transcription factor that eliminates synapses onto hippocampal CA1 neurons, but little is known of the neural activity patterns that lead to synapse elimination and whether this depends on MEF2. CA1 neurons fire in bursts at a theta frequency (3–8 Hz) in vivo during novel environment exploration and learning (*Colgin, 2013*) Using optogenetics to drive individual CA1 neurons in organotypic rat hippocampal slice cultures, *Goold and Nicoll (2010)* observed that chronic, 24 hr, bursting of CA1 neurons at 3 Hz (*Figure 1A*) resulted in elimination of excitatory synapses. We further explored this paradigm to determine whether shorter durations of postsynaptic bursting were sufficient to induce synaptic depression and elimination. Mouse organotypic hippocampal slice cultures were prepared, as described (*Pfeiffer et al., 2010*; *Wilkerson et al., 2014*), and a sparse population of CA1 neurons (<0.5%) was co-transfected with a mCherry-tagged Channelrhodopsin 2 (ChR2) and Cre using biolistics (*Figure 1A*). One to two weeks post-transfection, or equivalent postnatal days 15–21, slice cultures were exposed to *P*atterned *P*hoto*S*timulation (PPS; 50 ms pulses of blue light at 3 Hz) for increasing durations (1, 6 or 24 hr) which caused transfected, but not untransfected, CA1 neurons to fire in bursts of 1–3 action potentials (*Figure 1B*). 24–30 hr after the onset of PPS, we acquired dual, whole cell voltage clamp recordings from ChR2-transfected and neighboring untransfected neurons and measured the effect of PPS on excitatory synaptic function, evoked and spontaneous, or miniature (m) EPSCs. Transfected neurons exposed to 1 hr (brief) PPS displayed a robust (~50%) depression of evoked EPSC amplitude and mEPSC frequency relative to untransfected neurons (*Figure 1C*). In contrast, mEPSC amplitude and paired pulse facilitation of evoked EPSCs were unaffected (*Supplementary file 1*). As a control, transfection of ChR2, without PPS, had no effect on any measure of synaptic function (*Figure 1—figure supplement 1A*,

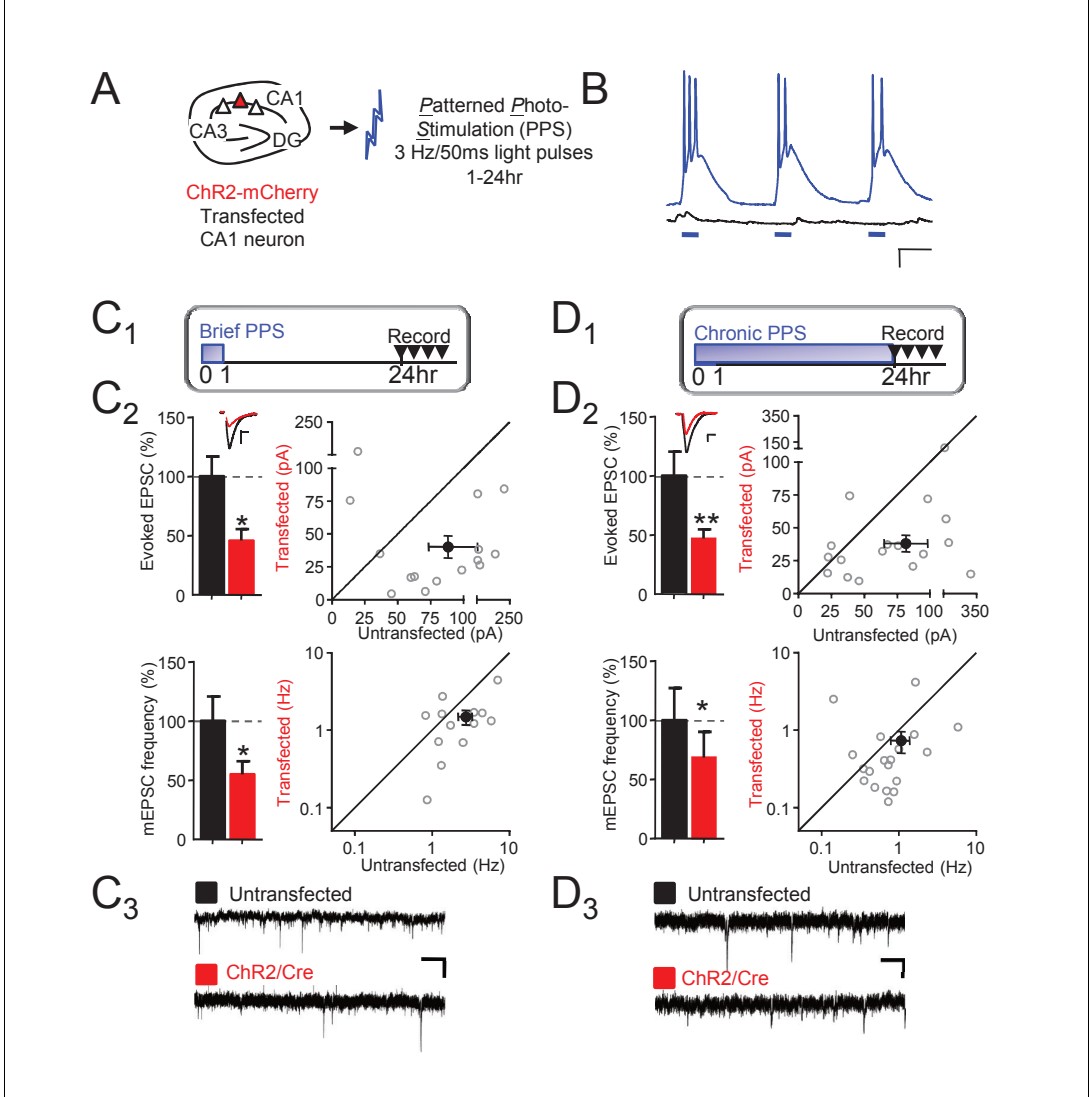

**Figure 1.** Theta frequency postsynaptic bursts of action potentials in CA1 neurons induce a long-term depression of excitatory synaptic transmission. (A) Experimental paradigm. Hippocampal slice cultures were prepared from wild-type (WT) mice, transfected with ChR-mCherry and Cre-mCherry and treated with patterned photostimulation (PPS; 3 Hz; 50 ms blue light pulses; 1 or 24 hr). (B) Patterned postsynaptic burst firing of ChR2 transfected (blue trace) or untransfected neurons induced by photostimulation (blue bars). Scale bar = 100 ms/ 10 mV (right panel). (C$_1$) Upper: Time course of brief PPS and recording. Simultaneous whole cell recordings from WT neurons transfected with ChR2-mCherry, Cre-mCherry, and MRE-GFP and neighboring untransfected neurons in cultures treated with brief PPS. Black fill triangles indicate the time window for recording. C$_2$ Left: Group averages of evoked EPSCs (upper) and mEPSC frequency (lower) from transfected (red fill) and untransfected neurons (black fill). Error bars in this and all figures represent SEM. Inset: Representative evoked EPSCs (scale = 10 ms/20 pA) from transfected (red) and untransfected (black) neurons. Right: Evoked EPSC amplitudes (upper) or mEPSC frequency (lower) from individual cell pairs (open circles). Transfected cell is plotted as a function of untransfected cell. Mean value is plotted as average ±SEM (filled circle). Diagonal line represents equality. C$_3$ Representative mEPSCs (scale = 500 ms/10 pA). (D$_{1-3}$). The same as C, except cultures were treated with chronic PPS. N = 13–19 cell pairs/condition. Statistic: Paired t-test.

DOI: https://doi.org/10.7554/eLife.26278.002

The following figure supplement is available for figure 1:

**Figure supplement 1.** Channelrhodopsin expression does not affect excitatory synaptic transmission.

DOI: https://doi.org/10.7554/eLife.26278.003

*Supplementary file 1*). Longer durations of PPS (6 or 24 hr; chronic PPS) also depressed evoked EPSCs and mEPSC frequency (measured 24 hr after PPS onset, *Figure 1D*), without affecting mEPSC amplitude or paired-pulse facilitation (*Supplementary file 1*). Chronic PPS or expression of MEF2VP16 in CA1 neurons causes a functional and structural elimination of synapses that manifests

as depressed EPSCs and mEPSC frequency without changes in mEPSC amplitude or paired-pulse facilitation (*Goold and Nicoll, 2010*; *Pfeiffer et al., 2010*; *Tsai et al., 2012*). Because brief PPS has similar effects on synaptic properties, it may also eliminate synapses.

## Brief periods of postsynaptic bursting silence excitatory synapses, whereas chronic bursting causes synapse elimination

To determine if brief PPS eliminates functional synapses, we measured NMDA receptor (NMDAR) mediated EPSCs. NMDARs colocalize with AMPA receptors (AMPAR) at excitatory synapses (*Kawashima et al., 2009*). Therefore, if brief PPS functionally eliminates synapses, we would predict a depression of both AMPAR and NMDAR-mediated EPSCs. To test this, we acquired paired recordings from ChR2-mCherry transfected and neighboring untransfected neurons 24 hr after the onset of brief PPS and measured AMPAR EPSCs and pharmacologically isolated NMDAR EPSCs (DNQX at +40 mV). Although brief PPS suppressed evoked AMPAR EPSCs, it had no effect on NMDAR EPSCs recorded 24–30 hr after the onset of PPS (*Figure 2A*). To determine if NMDAR EPSCs are initially depressed by brief PPS, but recover after PPS offset, we measured AMPA and NMDAR EPSCs at earlier time points after brief PPS onset, either 3–7 or 12–16 hr, and observed a selective depression of AMPAR, but not NMDAR EPSCs (*Figure 2B*; *Supplementary file 1*). Synapses with NMDARs, but not AMPARs are referred to as 'silent' synapses (*Kawashima et al., 2009*). Thus, brief PPS appears to silence excitatory synapses. In contrast, chronic PPS depressed both AMPAR and NMDAR EPSCs, recorded at either 24–30 or 48–54 hr (*Figure 2C,D*, *Supplementary file 1*) after the onset of chronic PPS, consistent with a functional synapse elimination.

To test the possibility that brief or chronic PPS differentially regulate synapse number, we measured the effects on dendritic spines, a structural correlate of excitatory synapses. CA1 neurons were co-transfected with myristoylated GFP (PA1-GFP) and ChR2-mCherry to allow visualization of dendritic spines with 2-photon microscopy. Brief PPS had no effect on spine density (*Figure 2E*), consistent with a functional synapse silencing. In contrast, chronic PPS reduced spine density on secondary apical dendrites, compared to transfected neurons without PPS (*Figure 2F*) which, together with our functional measurements indicate that chronic PPS eliminates synapses. As a control, chronic PPS of neurons transfected with GFP alone had no effect on spine number, indicating that decreased spine density was a result of light driven ChR2 activity (*Figure 2—figure supplement 1*). Together with our measurements of synaptic function, these data indicate that brief PPS silences excitatory synapses, whereas chronic PPS functionally depresses synaptic transmission and structurally eliminates spines, consistent with synapse elimination.

## Postsynaptic bursts of action potentials activate MEF2A/D-mediated transcription

Because of the proposed role of MEF2 in activity-induced synapse elimination (*Flavell et al., 2006*), we tested whether brief or chronic PPS activated MEF2-dependent transcription and if MEF2 is required for PPS-induced synapse silencing and elimination. To measure MEF2-regulated transcriptional activation in individual CA1 neurons, we co-transfected neurons with a MEF2 transcriptional reporter (MRE-GFP) and ChR2-mCherry. In unstimulated cultures, without PPS, MRE-GFP levels were low or undetectable, suggesting that basal activity levels were insufficient to drive endogenous MEF2. However, PPS stimulation for 1–24 hr robustly induced the MRE-GFP reporter (*Figure 3A,B*). To determine if endogenous MEF2A/D mediated PPS-induced MRE-GFP, we co-transfected CA1 neurons in slice cultures prepared from wildtype (WT) mice or mice with floxed alleles of *Mef2a* and *Mef2d* (*Mef2a/d*$^{fl/fl}$) (*Lyons et al., 1995*; *Akhtar et al., 2012*) with Cre-mCherry, ChR2-mCherry and MRE-GFP. Deletion or knockout (KO) of MEF2A/D blocked MRE-GFP induction in response to brief (1 hr) and chronic (24 hr) PPS (*Figure 3C,D*). MEF2A/D KO did not affect ChR2 expression, as assessed by blue light-induced currents (*Figure 3—figure supplement 1A*; *Supplementary file 1*). These results indicate that physiological action potential patterns activate MEF2A/D-driven transcriptional activation.

To determine if MEF2A/D is required for PPS-induced synapse silencing or elimination, we co-transfected CA1 neurons of *Mef2a/d*$^{fl/fl}$ slice cultures with Cre-mCherry and ChR2-mCherry and performed simultaneous recordings of neighboring transfected and untransfected neurons. MEF2A/D deletion, without PPS, had no effect on evoked or mEPSCs in unstimulated cultures (no PPS;

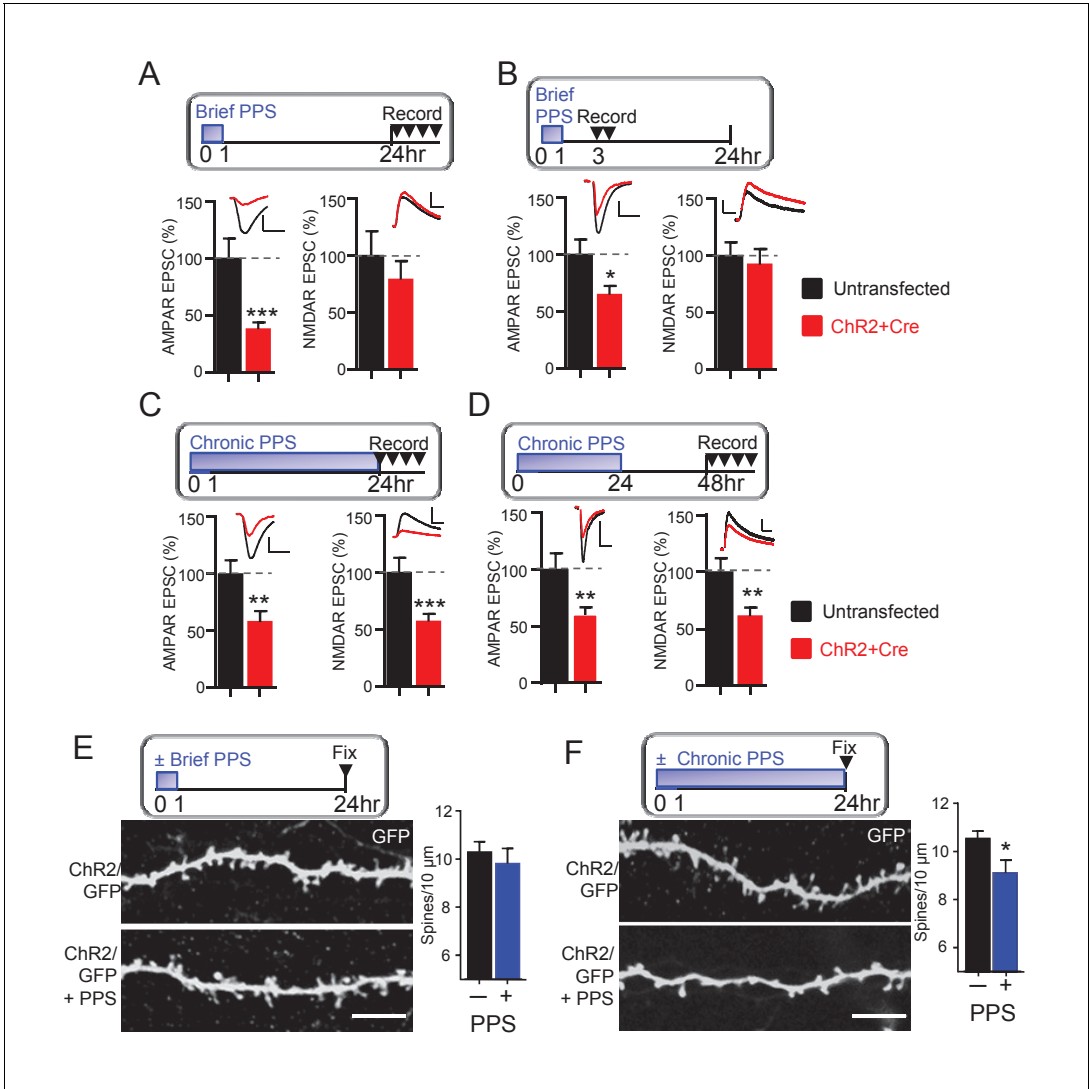

**Figure 2.** Brief PPS silences synapses, whereas chronic PPS induces functional and structural synapse elimination. (A) Upper: Time course of brief PPS and recording. Lower: Group average of evoked AMPAR (left) or NMDAR (right) mediated EPSC amplitudes from pairs of transfected and untransfected neurons after brief PPS. Inset: Representative evoked AMPAR and NMDAR EPSCs (scale = 20 ms/50 pA) from transfected (red) and untransfected (black) neurons. (B) The same as A, except slices were recorded 3–7 hr after PPS onset. (C) The same as A, except slices were treated with chronic (24 hr) PPS and recordings were performed 24–30 hr after PPS onset. (D) The same as C, except recordings were performed 48–54 hr after PPS onset. N = 13–22 cell pairs/condition. Statistic: Paired t-test. (E) Upper: Time course of brief PPS and fixation. Slice cultures, transfected with PA1-GFP and ChR2-mcherry, were subjected to brief PPS or not. Left: Representative images of secondary apical dendrites. Scale bar = 5 μm. Right: Group average of spine density on GFP + ChR2 transfected neurons with (blue bar) or without (black bar) PPS. (F) The same as D, except slice cultures were treated with chronic PPS or not. N = 13–20 cells/ condition. Statistic: Unpaired t-test. *p<0.05; **p<0.01; ***p<0.001.

DOI: https://doi.org/10.7554/eLife.26278.004

The following figure supplement is available for figure 2:

**Figure supplement 1.** Chronic PPS of CA1 neurons transfected with GFP alone has no effect on spine density.

DOI: https://doi.org/10.7554/eLife.26278.005

*Figure 4—figure supplement 1A*), but blocked the ability of brief PPS (1 hr) to depress EPSCs and mEPSC frequency (*Figure 4A,D*). These results, together with the results of the MRE-GFP reporter (*Figure 3*), suggest that basal activity in the slice cultures are insufficient to drive MEF2A/D transcriptional activity and synaptic depression, but elevations in postsynaptic action potentials drive MEF2A/D transcriptional activity and suppress synaptic function. In contrast to our results with brief PPS, MEF2A/D KO neurons had normal synaptic depression induced by chronic PPS (*Figure 4B,D*).

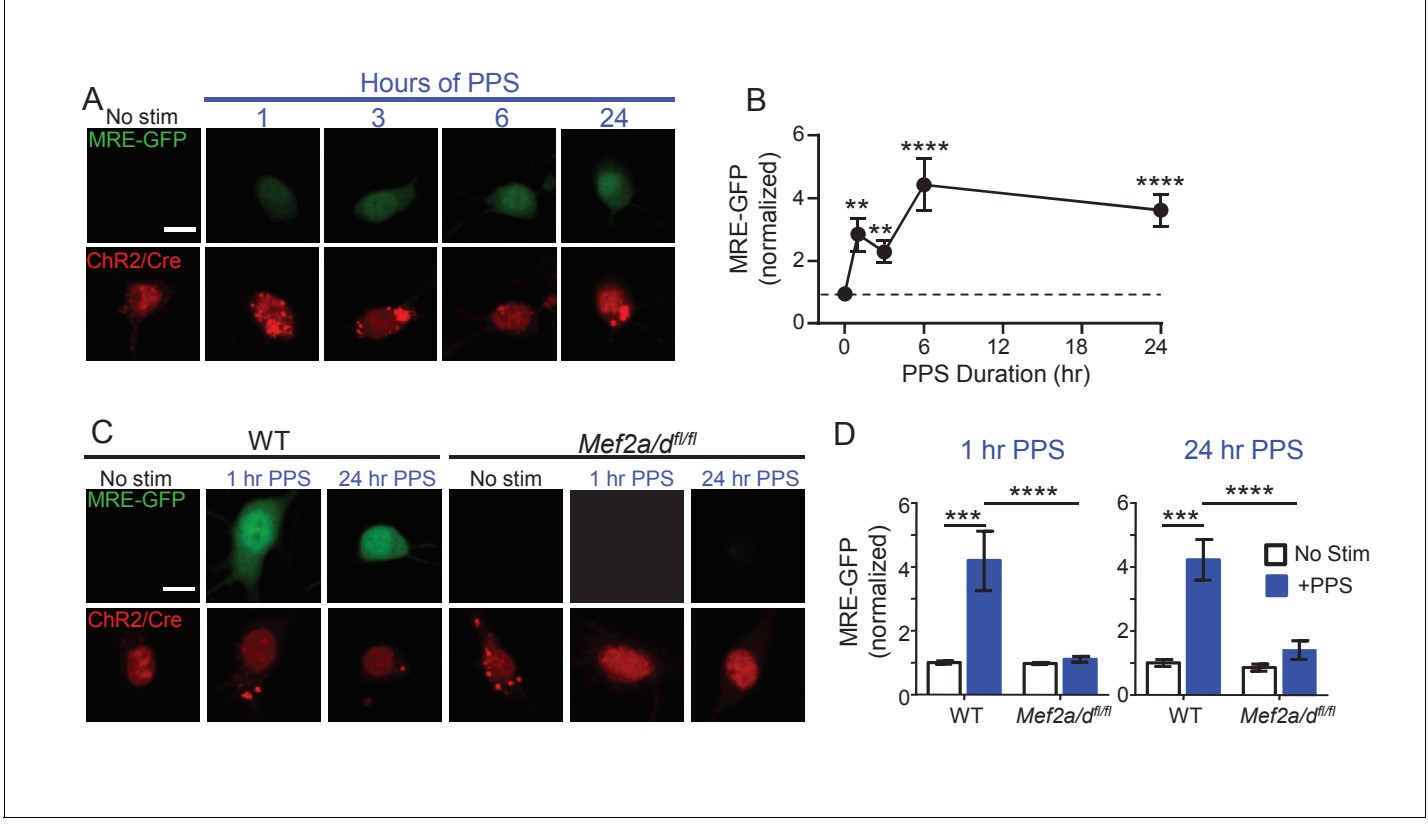

**Figure 3.** Brief and Chronic PPS activates MEF2A/D dependent transcription in individual CA1 neurons. (A) Representative images of MRE-GFP, ChR2-mCherry and Cre-mCherry in CA1 neurons treated with different durations of PPS or no stimulation (no stim). Neurons were imaged 24 hr after PPS onset. Scale bar = 10 μm. (B) Group average of normalized MRE-GFP expression in A. N = 26–52 cells/condition. Statistic: One-way ANOVA with Dunnet's multiple comparison. (C) Representative images for MRE-GFP, ChR2-mCherry and Cre-mCherry induced by 1 or 24 hr PPS in WT or *Mef2a/d*$^{fl/fl}$ neurons. Scale bar = 10 μm. (D) Group data of normalized MRE-GFP in C. N = 20–24 cells/condition. Statistic: Two-way ANOVA and Tukey's multiple comparison. **p<0.01; ***p<0.001; ****p<0.0001.

DOI: https://doi.org/10.7554/eLife.26278.006

The following figure supplement is available for figure 3:

**Figure supplement 1.** ChR2 function is not altered by MEF2A/D deletion.
DOI: https://doi.org/10.7554/eLife.26278.007

Compared to WT, 6 hr of PPS in MEF2A/D KO neurons induced only a trend towards a depressed evoked EPSC amplitude (*Figure 4C,D*, *Supplementary file 1*). These results support a specific role for MEF2A/D in activity-induced synapse silencing, but not functional synapse elimination.

The role of MEF2A/D in synapse silencing is observed when recordings are carried out a day after the cessation of brief PPS. To determine whether the differential role of MEF2A/D in synapse silencing vs. elimination is determined by the time of recording after cessation of PPS, we measured evoked AMPAR and NMDAR EPSCs a day after cessation of chronic PPS in WT and MEF2A/D KO neurons. Similar to WT, chronic PPS induced a persistent depression of AMPA and NMDAR EPSCs in neurons with MEF2A/D deletion, (*Figure 4—figure supplement 1B*, *Supplementary file 1*), supporting a specific role for MEF2A/D in synapse silencing in response to brief PPS.

### PPS-induced synapse silencing requires activation of L-type voltage-dependent Ca²⁺ channels and de novo transcription.

Brief PPS induces the MEF2 transcriptional reporter and silences synapses, both of which rely on MEF2A/D. These results suggest that postsynaptic bursting activates MEF2A/D-dependent transcription of gene targets that silence synapses. Neuronal depolarization drives MEF2A/D-mediated transcriptional activation by activating L-type voltage-gated Ca²⁺ channels (VGCC) and subsequent Ca²⁺ influx (*Flavell et al., 2006*). To determine if L-type VGCCs were required for MRE-GFP

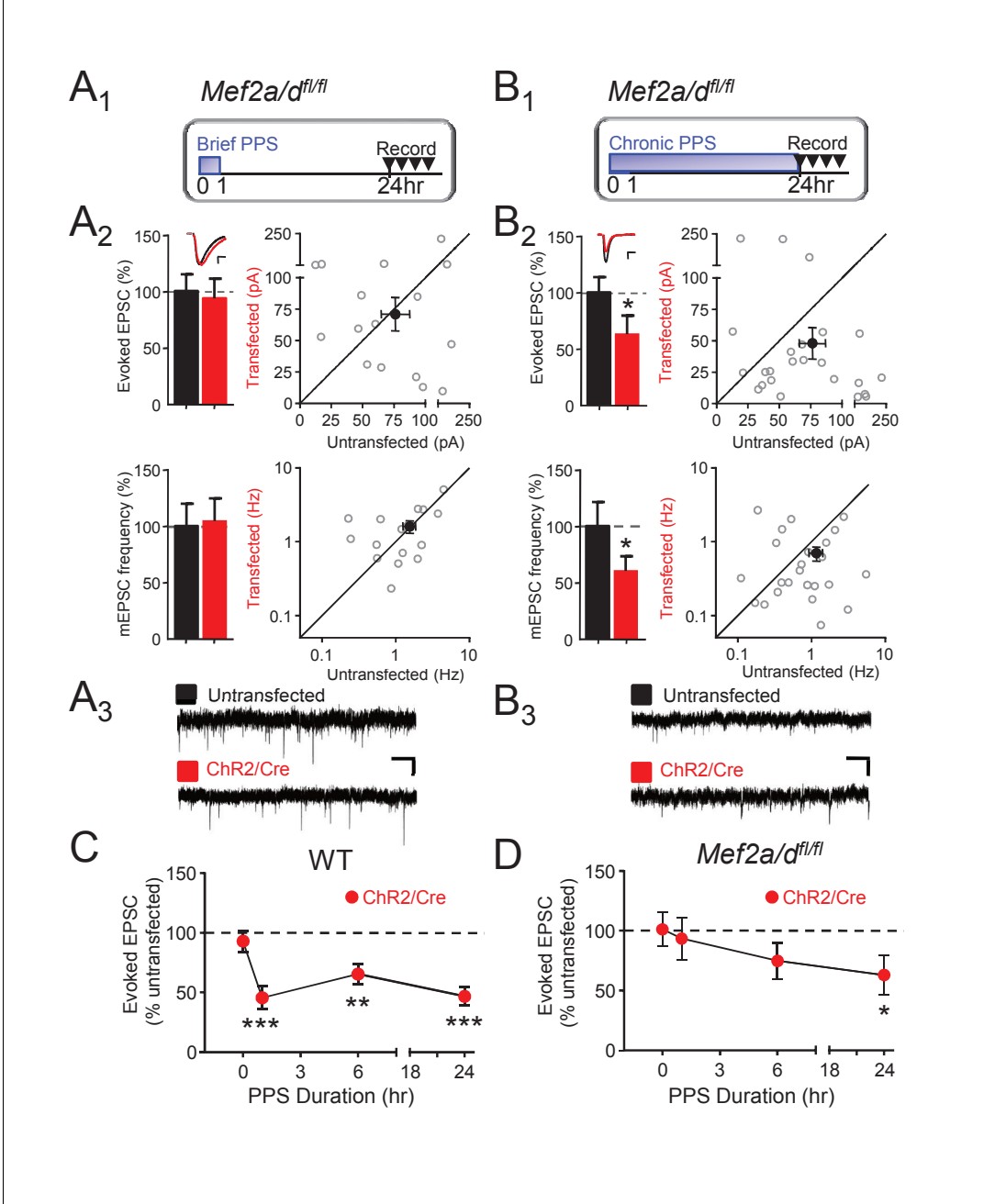

**Figure 4.** Postsynaptic MEF2A/D is necessary for synaptic depression induced by brief (1 hr), but not chronic (24 hr), PPS. (A₁). Time course of brief PPS and recording. (A₂) Left: Group averages of EPSC amplitudes (upper) or mEPSC frequency (lower) from simultaneous whole cell recordings from *Mef2a/d*<sup>fl/fl</sup> neurons transfected with ChR2-mCherry, Cre-mCherry, and MRE-GFP (red fill) and neighboring untransfected neurons (black fill) in cultures treated with brief PPS. Inset: Representative evoked EPSCs (scale = 10 ms/20 pA) from transfected (red) and untransfected (black) neurons. Right: Evoked EPSC amplitudes (upper) or mEPSC frequency (lower) from individual cell pairs (open circles). Transfected cell is plotted as a function of untransfected cell. Mean value is plotted as average ±SEM (filled circle). Diagonal line represents equality. (A₃) Representative mEPSCs (scale = 500 ms/10 pA). (B₁₋₃) The same as A except cultures were treated with chronic, 24 hr, PPS. N = 16–24 cell pairs/condition. Statistic: Paired t-test. (C) Group data of evoked EPSC amplitudes in transfected neurons (normalized) from WT slices either untreated (0 hr) or treated with increasing durations of PPS. (D) The same as C, except recordings are from *Mef2a/ d*<sup>fl/fl</sup> cultures. N = 13–24 cell pairs/condition. Wilcoxon signed rank with Bonferroni correction for multiple comparisons. *p<0.05; **p<0.01; ***p<0.001.

DOI: https://doi.org/10.7554/eLife.26278.008

*Figure 4 continued on next page*

*Figure 4 continued*

The following figure supplement is available for figure 4:

**Figure supplement 1.** Acute, postsynaptic deletion of *Mef2a/d* does not affect excitatory synaptic transmission and the persistence of chronic PPS-induced synapse depression.
DOI: https://doi.org/10.7554/eLife.26278.009

activation or synapse silencing in response to brief PPS, we pretreated slice cultures with nifedipine (20 µM) or vehicle before and during brief PPS. Under basal, or unstimulated conditions, nifedipine had no effect on MRE-GFP expression in comparison to vehicle-treated sister cultures. However, nifedipine blocked brief PPS-induced MRE-GFP (*Figure 5A,B*) as well as depression of evoked EPSCs and mEPSC frequency (*Figure 5C*) without affecting expression or function of ChR2 (*Supplementary file 1*). To determine if de novo transcription during PPS is necessary for synapse silencing, slice cultures were pretreated with the transcription inhibitor 5,6-Dichloro-1-beta-D-ribo-furanosylbenzimidazole (DRB; 160 µM) or vehicle (0.2% DMSO). As expected, DRB blocked PPS-induced expression of MRE-GFP (*Figure 5—figure supplement 1A,B*) and depression of mEPSC frequency (*Figure 5D*). We could not determine the ability of DRB to block PPS-induced depression of evoked EPSCs because DRB treatment alone, without PPS, depressed EPSCs in transfected neurons, in comparison to untransfected neurons (data not shown). Another transcription inhibitor with a distinct mode of action; actinomycin D (ActD; 1 µM; *Figure 5E*), also blocked brief PPS-induced depression of mEPSC frequency. We attempted to confirm the transcription dependence of chronic PPS-induced synaptic depression (*Goold and Nicoll, 2010*), but observed that 24 hr PPS +transcription inhibitors reduced the slice culture viability. Therefore, we tested the translation inhibitor, anisomycin (20 µM), which blocked chronic PPS induced depression of evoked and mEPSCs (*Figure 5—figure supplement 1C*). Together these results support a model where brief PPS activates L-type VGCCs to stimulate MEF2A/D-dependent transcription of gene targets that silence synapses.

## Brief PPS activates MEF2A/D-dependent Arc transcription, which is necessary, but not sufficient for synapse silencing

To identify MEF2-regulated genes induced by brief PPS which may play a role in synapse silencing, we infected primary dissociated, CA3-CA1 enriched, hippocampal cultures from WT or *Mef2a/d^{fl/fl}* pups with lentiviruses expressing ChR2-YFP and Cre-mCherry to drive neuron firing with PPS (*Figure 6—figure supplement 1A*). Cultures (DIV14) were harvested at 1, 3, 6, or 12 hr after PPS onset and we performed qPCR for MEF2A/D target genes, known to be induced by constitutively active MEF2 (MEF2-VP16) and/or tonic depolarization by high potassium: Activity-regulated cytoskeletal-associated protein, or *Arc*, *Homer1a*, Protocadherin 10 (*Pcdh10*), and Regulator of G-protein signaling (*Rgs2*) (*Flavell et al., 2006*; *Flavell et al., 2008*) (*Figure 6A* and *Figure 6—figure supplement 1B-E*). Some are candidate genes to mediate MEF2A/D-dependent synapse silencing. *Arc* is required for MEF2VP16-induced synapse elimination (*Wilkerson et al., 2014*) and stimulates endocytosis AMPA receptor (*Chowdhury et al., 2006*; *Shepherd et al., 2006*). *Homer1a* is necessary for activity-induced synaptic scaling (*Wang et al., 2006*), and *Pcdh10* is necessary for MEF2-VP16-induced synapse elimination and degradation of PSD-95 (*Tsai et al., 2012*). *Arc* was robustly induced (~20 fold) by the patterned postsynaptic bursting (brief PPS), at 3, 6 and 12 hr after PPS onset (*Figure 6A* and *Figure 6—figure supplement 1B*; n = 3–5 cultures). Although *Homer1a* levels were elevated (~3–5 fold), statistical significance was not reached (*Figure 6—figure supplement 1C*). Brief PPS did not induce *Rgs2* or *Pcdh10* (*Figure 6—figure supplement 1D,E*). To determine if PPS-induced *Arc* mRNA requires MEF2A/D, we repeated experiments in *Mef2a/d^{fl/fl}* cultures. Lentivirus-mediated Cre-mCherry decreased *Mef2a/d* RNA levels by 99% at DIV14, but did not alter basal *Arc* expression (data not shown). *Mef2a/d* deletion abolished PPS-induced *Arc* expression at all time points assayed (*Figure 6A*). Like WT neurons, PPS had no effect on *Homer1a* or *Rgs2* RNA levels in MEF2A/D KO cultures (data not shown). These results suggest physiological patterns of postsynaptic bursting selectively induce a subset of MEF2A/D target genes.

*Arc* is induced by brief PPS and relies on MEF2A/D, and may be required for synapse silencing. To test this idea, we examined PPS-induced synaptic depression in slice cultures from *Arc* KO mice

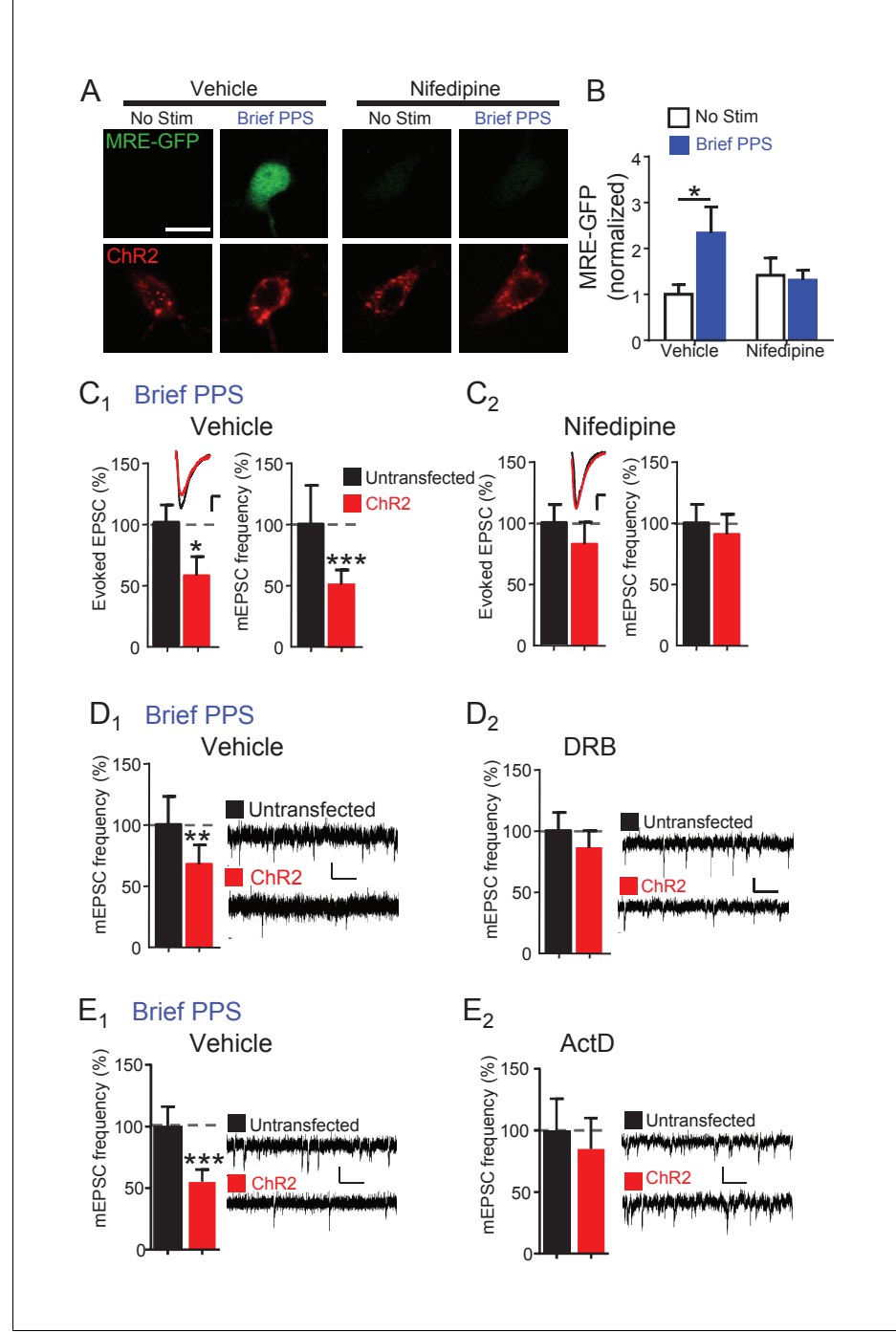

**Figure 5.** Synapse depression induced by brief PPS requires activation of L-type voltage-gated calcium channels and de novo transcription. (A) Representative images of MRE-GFP and ChR2-mCherry in CA1 neurons with or without brief PPS in vehicle (0.1% DMSO) or nifedipine (20 μM). Scale bar = 10 μm. (B) Group averages of MRE-GFP from experiments in A. Two-way ANOVA with Tukey's multiple comparison. N = 15–27 cells/condition. Statistic: Two-way ANOVA with Tukey's multiple comparison. (C) Group averages of evoked EPSC amplitude and mEPSC frequency from paired recordings of transfected and untransfected neurons subjected to brief PPS in the presence of vehicle ($C_1$) or nifedipine ($C_2$). Inset: Representative evoked EPSCs (scale = 10 ms/20 pA) from transfected (red) and untransfected (black) neurons. (D) Group averages of mEPSC frequency from transfected and untransfected neurons after brief PPS in vehicle ($D_1$; 0.2% DMSO) or DRB ($D_2$; 160 μM). Inset: Representative evoked EPSCs (scale = 10 ms/20 pA). (E) The same as D, except in actinomycin D (1 μM). Paired t-test. N = 12–19 cell pairs/condition. *p<0.05; **p<0.01; ***p<0.001.

*Figure 5 continued on next page*

*Figure 5 continued*

DOI: https://doi.org/10.7554/eLife.26278.010

The following figure supplement is available for figure 5:

**Figure supplement 1.** Effects of transcription and translation inhibitors on PPS-induced MRE-GFP and synaptic function.

DOI: https://doi.org/10.7554/eLife.26278.011

(*Wang et al., 2006*; *Wilkerson et al., 2014*). In support of our hypothesis, brief PPS failed to induce depression of AMPAR EPSCs or mEPSC frequency in *Arc* KO neurons in comparison to WT littermate controls (*Figure 6B*, *Supplementary file 1*). The deficit in activity-induced synapse silencing in *Arc* KO neurons could be due to a developmental or non-cell autonomous role for Arc, thus we attempted a rescue with postsynaptic expression of *Arc* cDNA from a constitutive CMV-promoter in *Arc* KO neurons. *Arc* cDNA expression in unstimulated cultures, without PPS, did not affect evoked EPSCs or mEPSCs (*Figure 6C1*, *Supplementary file 1*), indicating that expression of Arc alone is not sufficient to silence synapses. However, *Arc* cDNA expression in *Arc* KO neurons that received brief PPS rescued depression of evoked EPSCs and mEPSC frequency (*Figure 6C2*, *Supplementary file 1*). Together these results support a cell autonomous, homeostatic mechanism where postsynaptic bursts of activity activate a MEF2A/D-dependent induction of *Arc* which then functions to silence synapses onto that neuron.

To determine if other PPS-induced, MEF2A/D-dependent gene targets, in addition to *Arc*, are required for synapse silencing, we attempted to rescue PPS-induced synapse silencing by expressing *Arc* in MEF2A/D KO neurons. CA1 neurons of *Mef2a/d*$^{fl/fl}$ slice cultures were co-transfected with Cre-mCherry, ChR2-mCherry and *Arc* cDNA. Exogenous expression of Arc in MEF2A/D-KO neurons had no effect on mEPSCs in unstimulated cultures and failed to rescue PPS-induced synapse silencing (*Figure 6D*, *Supplementary file 1*). Therefore, exogenous Arc can rescue PPS-induced synapse silencing in *Arc* KO neurons, but not MEF2A/D KO neurons, implicating other activity- and MEF2A/D-dependent gene targets in synapse silencing, in addition to *Arc*.

Because MEF2A/D is dispensable for synapse elimination induced by chronic (24 hr) PPS (*Figure 4B*), Arc may be as well. To test this idea, we measured the effects of chronic PPS on synaptic function in the *Arc* KO. Chronic PPS failed to depress mEPSCs in *Arc* KO neurons and the depression of evoked EPSCs was quantitatively less in *Arc* KO neurons compared to WT littermates (*Figure 6E*, *Supplementary file 1*). Thus, Arc is essential for synaptic silencing and contributes quantitatively to synapse elimination in response to PPS. Other transcription factors that mediate activity-induced *Arc* (*Kawashima et al., 2009*) may contribute to synapse elimination or can compensate for loss of MEF2A/D.

## Discussion

Sensory experience, learning and neural circuit activity lead to elimination of excitatory synapses in cortical circuits, but little is known of the neural activity patterns that elicit activity-dependent synapse elimination or the molecular mechanisms of this process. Here we used optogenetics (patterned photostimulation; PPS) to drive firing of postsynaptic CA1 neurons in bursts at 3 Hz, similar to the theta frequency patterns of bursting observed in CA1 neurons during environmental exploration and sleep (*Colgin, 2013*). We observed two forms of synaptic depression that depended on the duration of postsynaptic bursting. Relatively brief periods of postsynaptic bursting (PPS; 1 hr) resulted in a selective depression of AMPAR-mediated synaptic transmission, or a silencing of excitatory synapses, whereas chronic (~24 hr) bursting activity depressed both AMPAR and NMDAR-mediated synaptic depression and reduced spine number, suggestive of a synapse elimination (*Goold and Nicoll, 2010*). Both synapse silencing and elimination depend on activity of L-type voltage-dependent Ca$^{2+}$ channels, transcriptional or translational activation (*Goold and Nicoll, 2010*) and *Arc*, but only synapse silencing requires the activity-dependent transcription factor MEF2A/D (*Figure 6—figure supplement 2*). Thus, different durations or levels of intracellular Ca2+ increases may engage different transcription factors and gene expression programs to homeostatically depress synaptic function through distinct mechanisms. These results reveal novel forms of activity

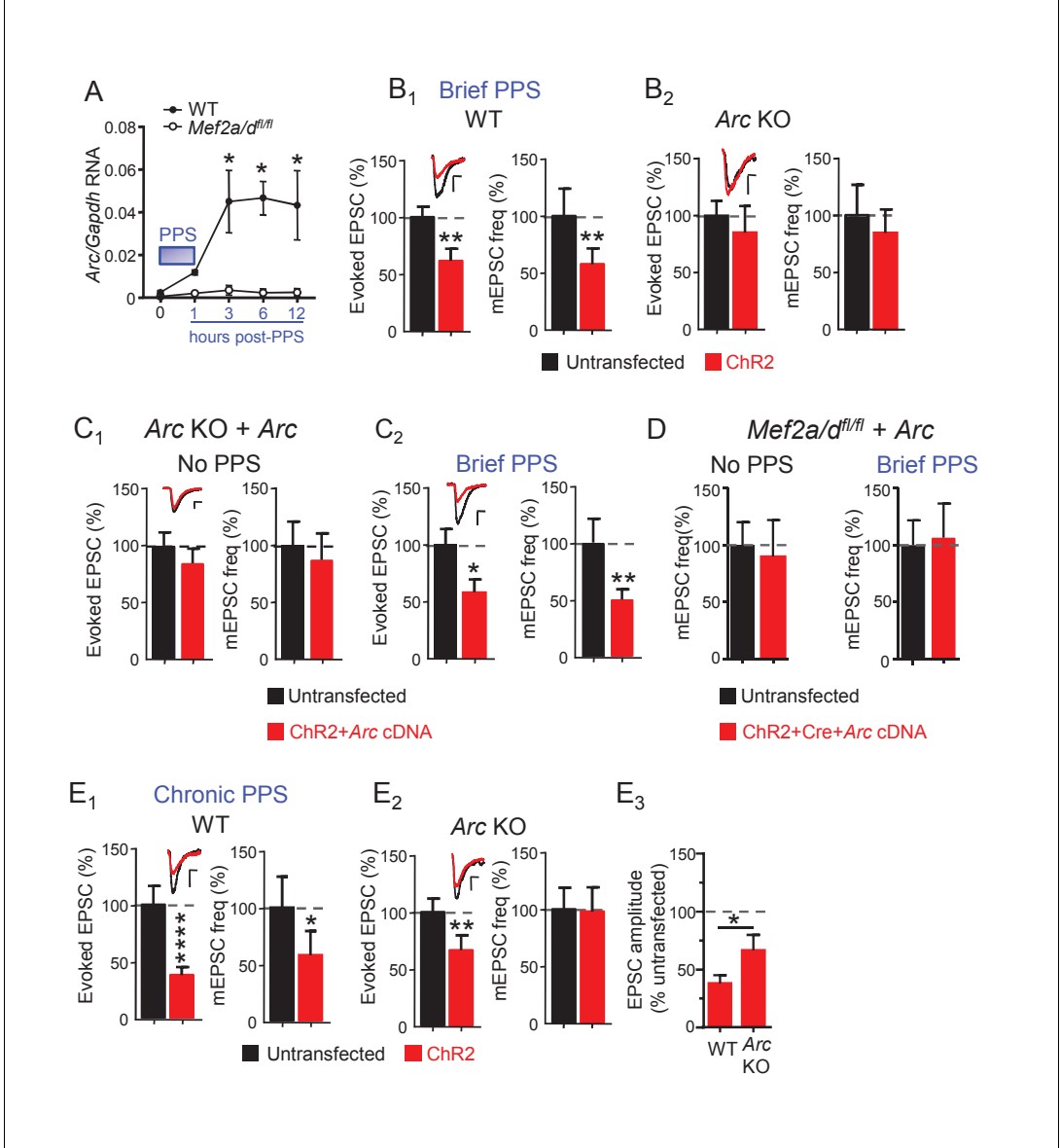

**Figure 6.** Brief PPS induces *Arc* which is necessary, but not sufficient, for synaptic depression. (**A**) *Arc* RNA expression after brief PPS in dissociated WT (closed circles) or *Mef2a/d*$^{fl/fl}$ (open circles) cultures transfected with Cre-mCherry and ChR2-YFP. N = 3–5 independent cultures. Statistic: Two-way ANOVA and Sidak's multiple comparison comparing WT vs. *Mef2a/d*$^{fl/fl}$. (**B**) Group averages of EPSC amplitude and mEPSC frequency from ChR2-transfected and untransfected neurons from WT (**B₁**) or Arc KO (**B₂**) slice cultures after brief PPS. Inset: Representative EPSCs (scale = 10 ms/20 pA) from transfected (red) and untransfected (black) neurons. (**C**) Group averages of EPSC amplitude and mEPSC frequency from untransfected Arc KO neurons and those cotransfected with ChR2 and Arc cDNA in cultures without (**C₁**) or with (**C₂**) brief PPS. Inset: Representative EPSCs (scale = 10 ms/20 pA). (**D**) Group averages of mEPSC frequency from untransfected *Mef2a/d*$^{fl/fl}$ neurons and those cotransfected with ChR2, Cre-mCherry and Arc cDNA without or with exposure to brief PPS. (**E**) Group averages of EPSC amplitude and mEPSC frequency from ChR2-transfected and untransfected neurons from WT (**E₁**) or Arc KO (**E₂**) cultures after chronic PPS. Inset: Representative EPSCs (scale = 10 ms/20 pA). N = 11–23 pairs/condition. Statistic: Paired t-test. (**E₃**) Group averages of EPSC amplitudes expressed as % untransfected neurons from WT and Arc KO cultures after chronic PPS. Independent t-test. *p<0.05; **p<0.01; ***p<0.001; ****p<0.0001.

DOI: https://doi.org/10.7554/eLife.26278.012

The following figure supplements are available for figure 6:

**Figure supplement 1.** Brief PPS robustly induces *Arc*.
DOI: https://doi.org/10.7554/eLife.26278.013

**Figure supplement 2.** Working model of synapse silencing and elimination by different durations of postsynaptic bursting.
DOI: https://doi.org/10.7554/eLife.26278.014

and transcription-dependent synaptic depression mechanisms and provide a model to understand how activity, and perhaps experience, regulates connectivity of cortical circuits.

## Network vs. cell-autonomous homeostatic synaptic depression

A well-studied form of homeostatic synaptic plasticity is induced by chronic changes in neural network activity, typically using pharmacological manipulations (*Wayman et al., 2008*). In contrast, much less is known of how cell autonomous activity of individual neurons homeostatically regulates their inputs (*Burrone et al., 2002*; *Ibata et al., 2008*). Indeed, *Goold and Nicoll (2010)* first demonstrated using chronic PPS that increases in cell autonomous activity depress and eliminate excitatory synaptic inputs. Here, we extend upon this study and demonstrate that distinct forms of homeostatic synaptic depression are induced depending on the duration of postsynaptic bursting. Brief (~1 hr) PPS caused a decrease in mEPSC frequency and evoked AMPAR-EPSCs without affecting NMDAR-EPSCs, dendritic spine number or paired-pulse facilitation of evoked EPSCs, a measure related to presynaptic release probability. Taken together, these results suggest that brief PPS causes a removal or inactivation of all AMPARs, at a subset of synapses, or creates silent synapses. Chronic PPS (~24 hr) also depressed the frequency of mEPSCs, but in contrast to brief PPS, depressed both evoked AMPA and NMDAR-mediated EPSCs and reduced spine number without affecting paired-pulse facilitation. If synapse silencing by brief PPS precedes or promotes removal of NMDARs and spines in response to chronic activity is unclear. Because brief or chronic PPS did not affect the average amplitude of mEPSCs, this suggests that synapses are silenced and eliminated across a range of synaptic strengths. However, measurements of synaptic protein markers are necessary to conclusively determine if synapses are eliminated by chronic PPS. The synaptic basis of this 'cell autonomous' homeostatic plasticity is distinct from the well characterized homeostatic 'scaling' down of mEPSC amplitudes observed with increased network activity, which results from removal or inactivation of a proportion of AMPARs from all synapses (*Wayman et al., 2008*; *Vitureira et al., 2012*). While homeostatic scaling maintains the relative synaptic weights of synaptic inputs onto a neuron (*Wayman et al., 2008*), cell autonomous homeostatic silencing would be expected to eliminate some synaptic inputs, while leaving others unchanged. Therefore, these two forms of homeostatic plasticity (cell autonomous vs. network) would have distinct effects on neural network connectivity and computation and may serve divergent functions.

## Endogenous MEF2A/D is required for cell autonomous, activity-induced synapse silencing

Brief and chronic PPS both require L-type $Ca^{2+}$ channel activity and transcriptional activation (*Goold and Nicoll, 2010*) for synapse silencing and elimination, but MEF2A/D is only required for brief PPS-induced synapse silencing. Consistently, brief PPS induced the MRE-GFP transcriptional reporter and target gene *Arc*, in a MEF2A/D-dependent manner. Functional synapse elimination in response to chronic PPS did not require MEF2A/D. This result was surprising because overexpression of a constitutively active MEF2, MEF2VP16, causes elimination of functional and structural synapses in hippocampal neurons (*Flavell et al., 2006*; *Pfeiffer et al., 2010*). Our data implicate postsynaptic MEF2A/D in activity-driven silencing of synapses. Consistent with this result, knockdown of MEF2A/D in developing cultured cortical neurons enhanced synaptic expression of GluA1, but not pre-or postsynaptic markers (*Elmer et al., 2013*), suggesting that endogenous MEF2A/D promotes synaptic removal or endocytosis of GluA1-containing AMPARs.

Surprisingly, postsynaptic deletion of MEF2A/D in CA1 neurons in slice culture did not affect baseline synaptic function. This result contrasts with the robust increase in synaptic transmission we have reported with postsynaptic expression of a constitutive transcriptional repressor form of MEF2, MEF2-Engrailed in CA1 neurons (*Pfeiffer et al., 2010*). MEF2A/D recruits transcriptional repressors to MEF2 target genes, such as class IIa HDACs, and deletion of MEF2A/D would be expected to prevent both repression and activity-induced activation of target genes (*Shalizi and Bonni, 2005*). Therefore, our results suggest that under basal activity conditions in slice cultures, MEF2A/D is not actively repressing transcription of genes that depress or eliminate synapses, but can be recruited by neural activity to induce transcription of gene targets, such as *Arc*, that silence synapses (*Figure 6—figure supplement 2*). Alternatively, steady-state levels of MEF2A/D and/or Arc may maintain the ability of CA1 neurons to depress in response to activity. Although acute blockade of

transcription blocks PPS-induced silencing, we do not know if PPS-induced transcriptional activity of MEF2A/D or induction of Arc induces synapse silencing.

Chronic PPS-induced synapse elimination relies on transcription and CaMKIV, a known regulator of activity-dependent transcription (*Wayman et al., 2008*; *Goold and Nicoll, 2010*), and, as we demonstrate, quantitatively on *Arc*. If or what activity-dependent transcription factors necessary for synapse elimination in response to chronic PPS are unknown. It is unlikely that compensation by MEF2C occurs with MEF2A/D deletion, because MEF2C is not expressed in CA1 pyramidal neurons postnatally (*Lyons et al., 1995*; *Kamme et al., 2003*) and MEF2A/D deletion fully prevented chronic PPS-induced MRE-GFP reporter expression. Important future experiments are to determine whether *Arc* is induced by chronic PPS in MEF2A/D KO neurons and if other activity-regulated transcription factors, such as CREB or SRF, can compensate or contribute to *Arc* induction and synapse elimination in the absence of MEF2A/D (*Kawashima et al., 2009*).

Previous studies have primarily used high KCl-mediated depolarization to induce MEF2A/D transcriptional activation and target genes (*Flavell et al., 2006*; *Flavell et al., 2008*). While these paradigms show some overlap with experience-induced genes in vivo, the physiologically-relevant neural activity patterns that drive gene expression is not known. CA1 neurons fire in bursts at a theta frequency (3–8 Hz) during exploratory or orienting behavior, as well as REM sleep (*Colgin, 2013*). Postsynaptic bursting at 3 Hz induced MEF2A/D dependent transcription of *Arc* and synaptic depression which may contribute to synaptic plasticity during these behaviors. Additional studies are needed to determine the relevant and optimal neural firing patterns that drive MEF2A/D-dependent transcription and synaptic plasticity and understand relevant in vivo activity conditions that drive these forms of synaptic plasticity. Surprisingly, brief PPS did not induce other MEF2A/D gene targets, such as *Pcdh10*, *Homer1a*, and *Rgs2*, which are induced by MEF2VP16 or high KCl depolarization (*Flavell et al., 2008*), suggesting that other activity patterns may also engage MEF2A/D to induce these genes. It is important to note that different genes may be induced with PPS of individual CA1 neurons in slice culture in comparison to dissociated CA3-CA1 cultures.

## Arc is necessary, but not sufficient, for synapse silencing and contributes to synapse elimination

Brief PPS robustly induced *Arc* in cultured hippocampal neurons and Arc was essential for synapse silencing in response to brief PPS. Arc promotes endocytosis of AMPARs (*Chowdhury et al., 2006*; *Shepherd et al., 2006*) and may do so through interactions of the AMPAR-interacting protein stargazin (TARPγ2) (*Penley et al., 2013*). However, postsynaptic exogenous expression of Arc was not sufficient to cause synaptic depression or elimination (*Wilkerson et al., 2014*), but rescued PPS-induced synaptic depression in Arc KO CA1 neurons. Therefore, other activity-induced, MEF2A/D dependent gene targets, in addition to *Arc*, are likely necessary to silence synapses. In support of this idea, Arc was unable to rescue PPS-induced synaptic depression in MEF2A/D KO neurons. Arc also contributed quantitatively to synapse elimination in response to chronic PPS. Consistent with this finding, Arc is required for MEF2VP16-induced synapse elimination (*Wilkerson et al., 2014*) and developmental and activity-induced elimination of climbing fiber axons onto cerebellar Purkinje neurons (*Mikuni et al., 2013*). Similar to what we observe in CA1, *Arc* overexpression is not sufficient eliminate climbing fibers (*Mikuni et al., 2013*). Arc-mediated synapse silencing may precede and target synapses for elimination during chronic activity increases. Alternatively, silencing and elimination may be distinct processes that both utilize Arc. Arc interacts with a number of synaptic proteins, including GluN2A, GluN2B, GKAP and WAVE1 (*Penley et al., 2013*), through which it may function to depress NMDAR function or spine number, independent of its effects on AMPARs. Because MEF2A/D is required for synapse silencing, but not elimination, this supports the idea that silencing is not necessary for elimination.

Alterations in synaptic pruning or cortical neurons, either during development, in response to experience or with aging are associated with a number of neuropsychiatric diseases, including autism, schizophrenia and Alzheimer's disease (*Tang et al., 2014*; *Sekar et al., 2016*). For example, the intellectual disability and autism-related gene, *FMR1*, mutated in Fragile X Syndrome, is necessary for developmental and experience-dependent synaptic pruning (*Patel et al., 2014*; *Nagaoka et al., 2016*) as well as pruning in response to MEF2VP16 (*Pfeiffer et al., 2010*). This work defines distinct stages of activity-regulated synapse elimination and provides a reduced and accessible preparation to determine the disease-related mechanisms in cortical synaptic pruning.

## Materials and methods

### Hippocampal slice cultures and transfection

Cultures were prepared from postnatal day (P) 6–7 WT mice, $Mef2a^{fl/fl}$ (RRID:MGI:5560661) and $Mef2d^{fl/fl}$ (RRID:MGI:5560659) mice (Akhtar et al., 2012; Zang et al., 2013), or Arc KO (RRID:MGI:3694763) mice (Wang et al., 2006) using previously published protocols (Stoppini et al., 1991; Pfeiffer et al., 2010). Organotypic hippocampal slice cultures were prepared from postnatal day (P) 6–7 mice of C57BL/6 mouse strain. Cultures were biolistically transfected at 3 DIV. Biolistic transfection and gold bullet preparation were performed with the Helios Gene Gun system (BioRad) according to the manufacturer's protocols (McAllister, 2004).

### Constructs

pLenti-CaMKIIa-hChR2(H134R)-mCherry-WPRE (Addgene plasmid # 20943; provided by Roger Nicoll) and pLenti-Synapsin-hChR2(H134R)-EYFP-WPRE (Addgene plasmid # 20945) were gifts from Karl Deisseroth (Zhang et al., 2007). FUGW_mCherry-NLS-Cre was a gift from Thomas Sudhof. MRE-GFP and Arc cDNA constructs have been previously described (Pfeiffer et al., 2010; Wilkerson et al., 2014).

### Patterned photostimulation and drug treatment

7–13 days after transfection, 6–8 hippocampal slice cultures consolidated on a culture plate insert in a 6-well plate were flashed by a collimated blue LED (470 nm) from Thorlabs (M470L3-C1 or M470L3-C5) inside a 35°C, 5% $CO_2$ humidified incubator. The duration of blue light flashing and post-flashing incubation time are indicated in text. The collimated light density at the location where slices were flashed was calibrated to 35 mW/mm$^2$, as measured by Fieldmax Top. LEDs were driven by a T-Cube LED Driver, 1200 mA Max Drive Current (Thorlabs), and the pattern (50 ms for each pulse at 3 Hz) of LED flash was controlled by a PC using Labview (National Instruments, Austin, TX; https://github.com/ColdP1228/LED-flashing-control-program [Gibson, 2017a]; copy archived at https://github.com/elifesciences-publications/LED-flashing-control-program. If drug treatment was required, vehicle or drug was added to slice culture media 20–60 min before the onset of photostimulation. Except for DRB, slices were maintained in vehicle or drug until used for electrophysiological assays or imaging. To maintain DRB in solution, 0.2%DMSO was required. This concentration of DMSO was detrimental to slices, thus, cultures were pre-treated with vehicle or DRB for 60 min, followed by a 1 hr PPS, then moved to fresh slice culture media 60 min after completion of the PPS protocol. Slices were assayed 24 hr post-PPS initiation.

### Imaging of MRE-GFP, the MEF2 transcriptional activity reporter

Transfected organotypic slice cultures (constructs indicated in the text) were pharmacologically treated (as indicated in the text), then subjected to photostimulation. 24 to 30 hr post-PPS, slice cultures were placed in a Warner chamber filled with warm Tyrode's solution containing (in mM): 150 NaCl, 4 KCl, 2 CaCl2, 2 MgCl2, 10 glucose, 10 HEPES. Single plane images (1024 × 1024 pixel resolution) were acquired using a Plan-Neofluar 63X/1.3 oil immersion objectives mounted on a Zeiss LSM 510 inverted confocal microscope. Quantification of green (MRE-GFP) and red (mCherry) somal fluorescence was performed using ImageJ software as previously described (Pulipparacharuvil et al., 2008; Pfeiffer et al., 2010). MRE-GFP expression profile was determined by normalizing GFP fluorescence intensity over background fluorescence intensity, and the resultant value was normalized to the average value from the control group without photostimulation. Background fluorescence was determined by a region equal to the region area of the analyzed neuron in the field of view adjacent to the neuron. mCherry fluorescence was not different between any group within studies, suggesting that pharmacological and/or genetic manipulations did not alter the expression of mCherry-tagged constructs (ChR2 or Cre). Two to three independent slice cultures (litters) were used in each imaging study. Statistical significance between groups was determined with a two-factor ANOVA (factor 1 = photostimulation, factor 2 = genotype or drug treatment) with Tukey's multiple comparison test.

## Electrophysiology

Simultaneous whole-cell recordings were obtained from CA1 pyramidal neurons in slice cultures visualized using IR-DIC and GFP/mCherry fluorescence to identify transfected and untransfected neurons (*Pfeiffer et al., 2010*). Recordings were made at ~30°C in a submersion chamber perfused at 2–3 ml/min with artificial cerebrospinal fluid (ACSF) containing (in mM): 119 NaCl, 2.5 KCl, 26 NaHCO3, 1 NaH2PO4, 11 D-Glucose, 4 CaCl2, 4 MgCl2, 0.1 picrotoxin, 0.002 2-chloro-adenosine; 0.1% DMSO pH ~7.28, 290–305 mOsm and saturated with 95% $O_2$/5%$CO_2$. For evoked EPSC (eEPSC) and mEPSC recordings, neurons were voltage clamped at −60 mV through whole cell recording pipettes (~4–7 MΩ) filled with an internal solution containing (in mM): 0.2 EGTA, 130 K-Gluconate, 6 KCl, 3 NaCl, 10 HEPES, 2 QX-314, 4 ATP-Mg, 0.4 GTP-Na, 14 phosphocreatine-Tris, pH 7.2 adjusted by KOH, 285–300 mOsm. To obtain isolated NMDAR mediated eEPSCs, the ASCF was supplemented with 20 µM DNQX and 20 µM glycine and the neuron was clamped at +40 mV. The internal pipette solution for NMDAR eEPSCs contained (in mM) 2.5 EGTA, 125 Cs-Gluconate, 6 CsCl, 3 NaCl, 10 HEPES, 10 sucrose, 10 TEA-Cl, 2 QX-314, 4 ATP-Mg, 0.4 GTP-Na, 14 phosphocreatine-Tris, pH7.2 adjusted by CsOH, 295 mOsm; for AMPAR eEPSCs acquired prior to NMDAR eEPSCs, the same internal solution was used, but ACSF was not supplemented with DNQX or glycine. For mEPSC measurements, the ACSF was supplemented with 1 µM TTX. Synaptic responses were evoked by single bipolar electrode or 2-conductor cluster electrode placed in stratum radiatum of area CA1 (along the Schaffer collaterals) 20–100 µm from the recorded neurons with monophasic current pulses (10–300 µA, 0.2–1 ms). Series and input resistance were measured in voltage clamp with a 400 ms, –10 mV step from a –60 mV holding potential (filtered at 30 kHz, sampled at 50 kHz). Cells were only used for analysis if the series resistance was less than 31 MΩ. Input resistances are included in supplementary tables. To assess the expression levels of functional ChR2, blue light pulses (470 nm) with maximum LED output or with a mercury arc lamp were applied after synaptic measurements and peak blue light-induced current ($I_{LED}$) are reported in the supplemental tables.

Synaptic currents were filtered at 2 kHz, acquired and digitized at 10 kHz on a PC using custom software (Labview; National Instruments, Austin, TX; https://github.com/ColdP1228/Custom-Data-Acquisition-Program- [*Gibson, 2017b*]; copy archived at https://github.com/elifesciences-publications/Custom-Data-Acquisition-Program-). mEPSCs were detected off-line using an automatic detection program (MiniAnalysis; Synaptosoft Inc, Decatur, Ga.) with a detection threshold set at a value greater than at least 5 standard deviations of the baseline noise, followed by a subsequent round of visual confirmation. The detection threshold remained constant for the duration of each experiment. For evoked EPSCs shown in figures the stimulation artifact has been digitally removed for clarity. Significant differences between transfected and untransfected neurons were determined by a paired t-test (normal distribution) or a Wilcoxon's signed rank test (non-normal distribution).

## Spine imaging

At 3 DIV, organotypic hippocampal slice culture were biolistically transfected with PA1-GFP, which expresses a myristoylated form of GFP to enhance filling of spines, and/or ChR2H134R-mCherry. At 8–13 d post-transfection (12–17 DIV), slices were subjected to photostimulation as indicated in the text. Slices were fixed in 2.5% PFA/4% sucrose for 1.5 hr, followed by permeabilization in 0.5% Triton X-100/10% normal donkey serum for 2 hr. Slices were incubated with 1° anti-GFP antibody (Aves Labs) at 4°C overnight, followed by incubation with 2° anti-chicken Alexa Fluor 488 antibody (Life Technologies) for 4 hr at room temperature. Secondary apical dendrites (150–200 µM from soma) of transfected CA1 neurons were imaged using a Zeiss LSM 780 2-photon laser scanning microscope. Images were obtained using an excitation wavelength of 920 nm and a 40 × 1.4 NA oil immersion objective. An interval of 0.3 µM and pixel resolution of 2048 × 2048 was used to acquire Z-stacks, generating images with pixel dimensions of 0.07 × 0.07 × 0.3 µM. For each neuron, 1–2 regions of interest were acquired. Images were analyzed using NeuronStudio, as previously described (*Rodriguez et al., 2006*; *Rodriguez et al., 2008*; *Wilkerson et al., 2014*). Imaging experiments were performed blind to treatment.

## Dissociated hippocampal neuron and glial cultures

Dissociated CA3-CA1 hippocampal cultures (dentate gyrus was discarded) were prepared from P0 mice using modified, previously published protocols (*Waung et al., 2008*). Briefly, dissected

hippocampi from P0 mice were trypsinized for 10 min then dissociated by trituration. After centrifugation, neurons were plated in Neurobasal A medium (Invitrogen) containing with B27 (2%; Invitrogen), 0.5 µM glutamine, and 1% fetal bovine serum (FBS) at a density of 450 neurons/mm2 on 35 mm dishes coated overnight with 50 µg/ml poly-D-lysine. One hr after plating, plates were washed 1 × with Neurobasal A supplemented with B-27 (Life Technologies) and 1% L-glutamine, and replaced with glial conditioned Neurobasal A medium (containing B27, glutamine and cytosine arabinoside (2 µM)) as described (*Niere et al., 2012*). At 1–2 DIV, cultures were infected with lentiviruses expressing SynI-ChR2-YFP and Cre-mCherry. Cultures were fed every 5 days by replacing 50% of the medium with glial conditioned medium. At 14–15 DIV, neurons were subjected to patterned photostimulation (PPS) as described in text. RNA was isolated 1, 3, 6, and 12 hr following onset of PPS.

Glial cultures were prepared from the neocortex of P0-P2 mouse pups and maintained in Neurobasal A containing 10% FBS and 50 µg/ml penicillin, 50 U/ml streptomycin, Sigma) for 3–4 weeks (*Viviani, 2006*). Neurobasal A media was conditioned for 48 hr, collected and stored at 4°C for no more than one week prior to addition to neuronal cultures.

## Virus preparation

HEK293T cells at 90% confluency were transfected with Rev, RRE, VSVg and lentiviral vector (ChR2-YFP or Cre-mCherry) using Lipofectamine 2000 (Invitrogen) according to the manufacturer's instructions. 12–18 hr after transfection, the cell culture media was replaced with Neurobasal A media containing B27. Media (where viruses were secreted) were harvested 48–60 hr after transfection, then filtered (0.45 µm filter, Millipore). Virus-containing media was used immediately or stored at 4°C for up to one week.

## RNA extraction and qPCR

At DIV14, dissociated cultures were subjected to PPS as indicated in the text, then RNA was extracted from treated dissociated cultures using TriZol reagent (Invitrogen), followed by purification using RNAeasy micro columns (Qiagen). Equal amounts of RNA were prepared for reverse transcription reactions using the Superscript II reverse transcriptase enzyme (Invitrogen). The efficiency of each primer set used in real-time quantitative PCR experiments was first tested on 10-fold serial dilutions of hippocampal cDNA to ensure that the primers promoted specific, exponential amplification of the target cDNA. Optimal primer sets for each gene were then used to assess the abundance of the reverse-transcribed mRNA in cDNA samples. PCR reactions were run in triplicate using iTaq universal SYBR green supermix. Each reaction was quantified using the $\Delta\Delta Ct$ method as previously described (*Tsankova et al., 2006*). Expression for each gene was normalized to GAPDH expression.

Primers used were: *Gapdh* forward: 5'-AGG TCG GTG TGA ACG GAT TTG-3'; *Gapdh* reverse: 5'-TGT AGA CCA TGT AGT TGA GGT CA-3'; *Arc* forward: 5'-AGC AGC AGA CCT GAC ATC CT-3'; *Arc* reverse: 5'-GGC TTG TCT TCA CCT TCA GC-3'; *Mef2a* forward: 5'-AAC CGA CAG GTT ACT TTT AC-3'; *Mef2a* reverse: 5'-TCT TAA CGT CTC AAC GAT AT-3'. *Rgs2* forward 5'-GCAAGGGTG TTGACGTTCTT-3'; *Rgs2* reverse 5'-TTTGGCACTCGTAACAGACG-3'. *Homer1a* forward 5'-TGA TTGCTGAATTGAATGTGTACC-3'; *Homer1a* reverse 5'-GAAGTCGCAGGAGAAGATG-3'. TaqMan probes (Life Technologies) were used for *Mef2d* – 00504931 and *Gapdh* – Mm99999915-g1.

## Statistics

All statistical analysis was performed in GraphPad Prism. For electrophysiology, the n equals the number of paired recordings. All experiments were performed in at least 3 independent slice cultures, prepared from different litters. Data from paired recordings were analyzed by a paired t-test or a nonparametric Wilcoxon signed rank paired test if the data did not pass a normality test (common with mEPSC frequency data). For analysis of % untransfected differences across genotypes (*Figure 6E3*), the ratio of EPSC amplitude in the transfected/untransfected neuron for each pair was log transformed, followed by an unpaired t-test. MRE-GFP expression and qPCR data was analyzed by one-way or two-way ANOVA followed by multiple comparisons as indicated in the figure legends. $*p<0.05$; $**p<0.01$; $***p<0.05$, $****p<0.001$. All grouped data are plotted as mean ±standard error of the mean.

## Acknowledgements

This work was supported by grants from Simons Foundation (#206919 to KMH), and NIH (R01-HD052731; KMH; R01-HD056370; JRG). We would like to thank Maria Diosdado, Nicole Cabalo and Darya Gonzalez for technical assistance.

## Additional information

### Funding

| Funder | Author |
| --- | --- |
| Simons Foundation | Chia-Wei Chang |
| National Institutes of Health | Chia-Wei Chang<br>Julia Wilkerson<br>Carly Hale<br>Jay R Gibson<br>Kimberly M Huber |

The funders had no role in study design, data collection and interpretation, or the decision to submit the work for publication.

### Author contributions

Chia-Wei Chang, Conceptualization, Data curation, Formal analysis, Validation, Investigation, Methodology, Writing—original draft, Writing—review and editing; Julia R Wilkerson, Data curation, Formal analysis, Validation, Investigation, Methodology, Writing—review and editing; Carly F Hale, Data curation, Formal analysis, Investigation, Methodology, Writing—review and editing; Jay R Gibson, Resources, Software, Methodology; Kimberly M Huber, Conceptualization, Resources, Data curation, Supervision, Funding acquisition, Project administration, Writing—review and editing

### Author ORCIDs

Chia-Wei Chang ⓘ http://orcid.org/0000-0002-1011-6870
Kimberly M Huber ⓘ http://orcid.org/0000-0002-7479-0661

### Ethics

Animal experimentation: All experimental protocols involving mice were performed in accordance with the guidelines and regulations set forth by the Institutional Animal Care and Use Committee at The University of Texas Southwestern Medical Center. The IACUC Protocol Number is 2017–101986.

### Decision letter and Author response

Decision letter https://doi.org/10.7554/eLife.26278.017
Author response https://doi.org/10.7554/eLife.26278.018

## Additional files

### Supplementary files

• Supplementary file 1. Raw electrophysiological measurements in untransfected (U) or transfected (T) hippocampal CA1 neurons.
DOI: https://doi.org/10.7554/eLife.26278.015

• Transparent reporting form
DOI: https://doi.org/10.7554/eLife.26278.016

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
