## [Decision Letter]

Thank you for submitting your article "Distinct stages of synapse elimination are induced by burst firing of CA1 neurons and differentially require MEF2A/D" for consideration by *eLife*. Your article has been favorably evaluated by Eve Marder (Senior Editor) and three reviewers, one of whom is a member of our Board of Reviewing Editors. The reviewers have opted to remain anonymous.

The reviewers have discussed the reviews with one another and the Reviewing Editor has drafted this decision to help you prepare a revised submission.

Summary:

In this manuscript the authors use channelrhodopsin to deliver specific patterns of activity to CA1 pyramidal neurons in organotypic mouse hippocampal slices in order to gain new insights into the transcriptional mechanisms of synaptic depression and elimination. This study combines two previous lines of investigation – one of which previously showed that prolonged low frequency ChR2-mediated stimulation of hippocampal neurons was sufficient to drive transcription-dependent synapse elimination and the other which has implicated the transcription factor *MEF2* in activity-dependent synapse elimination. Here the authors first show that both short and long-term delivery of patterned ChR2-mediated neuronal firing can induce *MEF2*-dependent transcription, then they use knockouts of MEF2A/D and its target gene *Arc* to determine the requirements for this transcriptional signaling pathway in excitatory synaptic depression and elimination. These data reveal that both MEF2A/D and *Arc* are required for activity-induced synaptic depression, but only *Arc* and not MEF2A/D are required for activity-induced synaptic elimination.

As the authors state in their Introduction, nothing is known of the physiological patterns of stimulation that are sufficient to activate *MEF2*-dependent transcription, thus the report of this finding alone is highly novel. However this paper is particularly important because it takes a further step toward combining the extensive molecular knowledge of activity-inducible transcription factors and placing it in the context of physiologically relevant effects on synapse plasticity.

Essential revisions:

There are a number of points that should be addressed with respect to the writing of the manuscript. Experimentally, the only significant weaknesses are at the 24hr time point after PPS, where adding findings will more firmly support the model proposed.

Writing:

1) The authors need to be more careful in their wording so as to be more accurate in their descriptions of the data. As an example, spine loss is not necessarily synapse loss, and so it should not be stated as such. In some places, this is appropriately stated, but in others (e.g. subsection “Brief periods of postsynaptic bursting silence excitatory synapses, whereas chronic bursting causes synapse elimination”), it is not. In an example of the second point, the authors state that PPS induced a MEF2A/D-dependent induction of *Ar*c that was necessary but not sufficient for synapse silencing and elimination. Although the data are compelling suggesting that *Arc* is necessary for depression (neurons from the *Arc* KO mouse failed to show synaptic depression after brief PPS, and neurons transfected with *Arc* cDNA can rescue PPS-induced synapse silencing in dissociated cultures, Figure 5), the data also show that chronic PPS can induce a significant decrease in evoked EPSCs (Figure 5E2) in the *Arc* KO mouse slices, demonstrating that *Arc* is, in fact, not necessary for depression, at least with chronic PPS. The authors did find that the depression in the KO was significantly different from the WT (Figure 5E3), however the decrease is similar to other decreases in evoked EPSC's that are considered depressed (e.g. 5B1).

2) The authors repeatedly state the importance of defining the relationship between physiological activity patterns and both *MEF2* activation and synaptic refinement; yet they only test a single pattern – albeit a physiologically relevant one – throughout the course of the study. This raises a number of questions that could be addressed experimentally or, at least, raised as Discussion points. 1) Is there anything significant about 3 Hz burst firing with respect to *MEF2* activation and synapse silencing/elimination? Would other patterns of burst firing (e.g. high theta, or γ) yield different outcomes?; 2) Is burst firing of the postsynaptic neuron required? Related, How reliably does the specified 50 ms blue light pulse evoke a burst of action potentials in the postsynaptic cell?; 3) Is it only the duration of the 3 Hz postsynaptic activity that differentially engages pathways for synapse silencing or elimination, or does the total amount of activity play a role? In other words, could high frequency activity (~15-20 Hz, allowing for a corresponding shortening of the light pulse duration?) administered for only 1 hr possibly engage the same transcriptional networks that promote synapse elimination in response to 24 hrs of 3 Hz stimulation? Obviously, the parameter space for patterned stimulation protocols is endless, but a discussion of these issues is warranted.

Experiments:

1) The authors performed the experiment in Figure 5 to determine if depressed NMDAR EPSC amplitudes might be detected nearer to cessation of the brief PPS. Could 12 hours post-PPS still be too late to detect the effect? The spread of the data in Figure 5 suggests this could be a possibility. Alternatively, would depressed NMDAR EPSC amplitudes manifest 24 hrs following the cessation of chronic PPS?

2) The authors should show if *Arc* is still induced by 24hrs of PPS since that is the stimulus used for the synaptic elimination study. Also it is important to know if *Arc* is induced in the absence of MEF2A/D after 24 hrs of PPS since that would be assumed from the data.

---

## [Author Response]

Essential revisions:There are a number of points that should be addressed with respect to the writing of the manuscript. Experimentally, the only significant weaknesses are at the 24hr time point after PPS, where adding findings will more firmly support the model proposed.Writing:1) The authors need to be more careful in their wording so as to be more accurate in their descriptions of the data. As an example, spine loss is not necessarily synapse loss, and so it should not be stated as such.

Thank you. We agree with the reviewer that spine loss does not equal synapse loss. We revised the wording to indicate that based on a combination of measures, depression of evoked AMPA and NMDAR EPSCs and mEPSC frequency, but not mEPSC amplitude, and spine loss, suggests that chronic PPS causes an elimination of synapses. And also added to the Discussion “measurements of synaptic protein markers are necessary to determine if synapses are eliminated by chronic PPS.”

In some places, this is appropriately stated, but in others (e.g. subsection “Brief periods of postsynaptic bursting silence excitatory synapses, whereas chronic bursting causes synapse elimination”), it is not. In an example of the second point, the authors state that PPS induced a MEF2A/D-dependent induction of Arc that was necessary but not sufficient for synapse silencing and elimination. Although the data are compelling suggesting that Arc is necessary for depression (neurons from the Arc KO mouse failed to show synaptic depression after brief PPS, and neurons transfected with Arc cDNA can rescue PPS-induced synapse silencing in dissociated cultures, Figure 5), the data also show that chronic PPS can induce a significant decrease in evoked EPSCs (Figure 5E2) in the Arc KO mouse slices, demonstrating that Arc is, in fact, not necessary for depression, at least with chronic PPS. The authors did find that the depression in the KO was significantly different from the WT (Figure 5E3), however the decrease is similar to other decreases in evoked EPSC's that are considered depressed (e.g. 5B1).

Thank you, we have corrected our wording to indicate that “*Arc is* necessary, but not sufficient, for synapse silencing and contributes quantitatively to synapse elimination.”

2) The authors repeatedly state the importance of defining the relationship between physiological activity patterns and both MEF2 activation and synaptic refinement; yet they only test a single pattern – albeit a physiologically relevant one – throughout the course of the study. This raises a number of questions that could be addressed experimentally or, at least, raised as Discussion points. 1) Is there anything significant about 3 Hz burst firing with respect to MEF2 activation and synapse silencing/elimination? Would other patterns of burst firing (e.g. high theta, or γ) yield different outcomes?; 2) Is burst firing of the postsynaptic neuron required? Related, How reliably does the specified 50 ms blue light pulse evoke a burst of action potentials in the postsynaptic cell?; 3) Is it only the duration of the 3 Hz postsynaptic activity that differentially engages pathways for synapse silencing or elimination, or does the total amount of activity play a role? In other words, could high frequency activity (~15-20 Hz, allowing for a corresponding shortening of the light pulse duration?) administered for only 1 hr possibly engage the same transcriptional networks that promote synapse elimination in response to 24 hrs of 3 Hz stimulation? Obviously, the parameter space for patterned stimulation protocols is endless, but a discussion of these issues is warranted.

These are all very interesting questions and is a future direction of this work. At this time, we have only tested 3 Hz, 50 msec light pulse protocol, and the effects of different durations of stimulation. In this revision, we included, as requested, additional results on the time course and persistence of synaptic changes after cessation of brief or chronic PPS (new Figure 2) and the dependence on MEF2A/D (new Figure 4—figure supplement 1). We observed in our early experiments when we optimized the intensity blue light pulses, we observed that neurons fired in bursts (2 or more action potentials) in ~85% of the pulses and in single spikes ~15% of pulses. Because L-type Ca^2+^ channels, are required for the MEF2A/D transcriptional activation (Flavell et al., 2006) and synaptic depression, and these typically require strong depolarization, we would anticipate that action potentials and a burst or prolonged depolarization (~80-100 msec) would be required (Kapur et al., 1998). We added in the Discussion”Additional studies are needed to determine the relevant and optimal neural firing patterns that drive MEF2A/D-dependent transcription and synaptic plasticity and understand relevant in vivo activity conditions that drive these forms of synaptic plasticity.”

Experiments:1) The authors performed the experiment in Figure 5 to determine if depressed NMDAR EPSC amplitudes might be detected nearer to cessation of the brief PPS. Could 12 hours post-PPS still be too late to detect the effect? The spread of the data in Figure 5 suggests this could be a possibility. Alternatively, would depressed NMDAR EPSC amplitudes manifest 24 hrs following the cessation of chronic PPS?

We assume the reviewer is referring to the original Figure 3 and these are excellent points. We performed several additional experiments to determine whether the depression of NMDARs EPSCs was determined by the duration of PPS (brief vs. chronic) or when, after PPS, we performed the recordings. As suggested by the reviewer, we recorded evoked AMPAR and NMDAR EPSCs, at earlier time points after brief PPS, 3-7 hours (new Figure 2). Similar to that observed with recordings at 12-16 or 24-28 hours after brief PPS, evoked AMPAR EPSCs were selectively depressed and NMDAR EPSCs were unchanged.

These results strengthen our original conclusions and indicate that brief PPS silences synapses within 3 hrs and this change persists for >24 hr.

As requested by the reviewer, we performed the converse experiment with chronic PPS, and waited >24 hours after cessation of chronic PPS to record evoked AMPAR and NMDAR EPSCs in both WT (new Figure 2) and MEF2A/D deleted neurons (new Figure 4—figure supplement 1). As observed in experiments recording immediately after chronic PPS, we observed persistent (>24 hr) depression of both AMPAR and NMDAR EPSCs in WT (new Figure 2) and MEF2A/D deleted neurons (new Figure 4—figure supplement 1). This result was also previously observed in rat hippocampal slice cultures (Goold and Nicoll, 2010). These results strengthen our original conclusions and indicate that the duration of PPS determines whether synapses are silenced or functionally eliminated and silencing selectively utilizes MEF2A/D.

2) The authors should show if Arc is still induced by 24hrs of PPS since that is the stimulus used for the synaptic elimination study. Also it is important to know if Arc is induced in the absence of MEF2A/D after 24 hrs of PPS since that would be assumed from the data.

We agree with the reviewer and attempted to perform this experiment. Unfortunately, we have had technical problems with the health of our dissociated neuron cultures for the past few months, perhaps caused by lot of changes in reagents, that has prevented us from observing robust, reliable induction of *Arc* with chronic PPS. We are willing to continue to troubleshoot these experiments, but practical interests of having a publication in a timely manner (Dr. Chang going to a postdoc, Dr. Huber’s grant renewal) dictate that we not further delay resubmission of the paper. We hope the reviewers will consider our sincere attempts to do this experiment and allow us to add to the Discussion as an alternative “Important future experiments are to determine whether *Arc* is induced by chronic PPS in MEF2A/D KO neurons and if other activity-regulated transcription factors, such as CREB or SRF, can compensate or contribute to *Arc* induction and synapse elimination in the absence of MEF2A/D (Kawashima et al., 2009).”